The impact of smog on the concentration of particulate matter in the antelope house in the Silesian zoological garden

Pawlak Krzysztof 1 rzpawlak@cyfronet.pl
http://orcid.org/0000-0002-2746-2534 Nieckarz Zenon 2 3
1 Agricultural University of Cracow, Department of Zoology and Animal Welfare , Faculty of Animal Science ; Cracow, Poland
2 Physical Education Academy in Cracow, Department of Muscle Physiology, Faculty of Rehabilitation , Cracow , Poland
3 Jagiellonian University Cracow, Experimental Computer Physics Department, Institute of Physics , Cracow , Poland
Phillips Gavin
Electronic publication date: 2020 May 28
Publication date: 2020
Volume: 8
Electronic Location ID: e9191
Received 2019 Nov 6; Accepted 2020 Apr 23
Copyright: © 2020 Pawlak and Nieckarz
Copyright year: 2020
Copyright holder: Pawlak and Nieckarz
License: This is an open access article distributed under the terms of the Creative Commons Attribution License, which permits unrestricted use, distribution, reproduction and adaptation in any medium and for any purpose provided that it is properly attributed. For attribution, the original author(s), title, publication source (PeerJ) and either DOI or URL of the article must be cited.
License URL: https://creativecommons.org/licenses/by/4.0/

Keywords: Buildings, Animals, Smog, Particulate matter

Funding: Faculty of Animal Science, University of Agriculture in Krakow, Poland 215/DZ06 This research was financially supported by the statutory activity 215/DZ06 of the Faculty of Animal Science, University of Agriculture in Krakow, Poland. The funders had no role in study design, data collection and analysis, decision to publish, or preparation of the manuscript.

==============================
Persistent negligence in the field of environmental protection in Poland as well as strong dependance of the energy sector on the fossil fuels have led to serious pollution of the air with particulate matter, which at high concentrations is capable of penetrating into the buildings. The aim of this study is to assess the impact of particulate airborne pollution on the concentration of particulate matter inside the antelope house in the Silesian Zoological Garden located within the Upper Silesia in Poland. The research was conducted from February to May in 2018. The records taken in the research period show that the concentration of PM10 exceeded the level of 50 µg/m3 outside the building during 26 days and 11 days when it comes to the concentration of particulate matter inside the antelope house. The quantity of particulate matter in the antelope house is strongly correlated with the concentration of the particles in the air. Despite fitting existing ventilation system with a filter that reduces the dust level by 60% during the highest level of smog, particulate matter concentration in the antelope house exceeded acceptable limit for PM10 more than twofold. Particle size-fraction analysis revealed that as much as 85% of the particles detected in the studied compartment constitute PM2.5.

Introduction

Sulfurous smog, also called London smog, is produced in humid air that is strongly polluted with acidic gases, mainly with sulfur dioxide (SO2), carbon dioxide (CO2), nitric oxide (NO) and particulate matter (Polivka, 2018; Poulopoulos, 2016). Particles with the diameter exceeding 2.5 μm that are present in smog are generated as a result of mechanical actions, such as grinding or crushing various types of materials (WHO, 2006). These materials either occur naturally—mineral dust, sea salt, or represent anthropogenic substances—produced as a consequence of tire and brake abrasion. Fine particulate matter particles, with the diameter below 2.5 μm, is formed mainly through the burning of fossil fuels in the utilities sector (Seinfeld & Pandis, 2006). Particles present in smog contain metal oxides, acidic condensates, heavy and transition metals (Wróbel, Rokita & Maenhaut, 2000), sulfates and nitrates, as well as elemental and organic carbon (Routledge & Ayres, 2005).

Health and the quality of life, both human and animal, is tightly connected with the condition of the environment. It is assumed that airborne particulate pollution constitutes one of ten most significant threats to global health (Lim et al., 2012). The research into the impact of smog on living organisms revealed that particle pollutants increase the risk of developing asthma and allergy (Chen et al., 2018), cause arrhythmia (Hazari et al., 2009), lung cancer (Pope et al., 2002), chronic obstructive lung disease (Zanobetti, Bind & Schwartz, 2008), bronchitis (King et al., 2018), as well as decrease fertility (Selevan et al., 2000; Yi, Wei & Fan, 2017) and constitute a risk factor for abnormal fetal development (Perera et al., 2002).

Prolonged ignorance as to the environmental conservation in Poland as well as heavy reliance of the energy industry on fossil fuels resulted in substantial increase in the particulate air pollution (http://www.gios.gov.pl/stansrodowiska/gios/pokaz_artykul/en/front/stanwpolsce/srodowisko_i_zdrowie/zanieczyszczenia_powietrza/; Tainio et al., 2013) This problem is especially serious in the Upper Silesian Agglomeration where acceptable 24-h PM10 concentration is exceeded 67–102 days per 1 year (Report, 2018).

The research conducted by Massey et al. (2009) and Chen et al. (2016) demonstrated that when air pollution is heavy particles are capable of migrating into the buildings. This problem becomes even more serious in the case of facilities intended for animals where particles penetrate as a result of permeability of the construction as well as through supply ventilation, which constitutes a regular and indispensable element of proper equipment in animal facilities. During winter, when the incidence of smog greatly increases, flow rate for the air exchanged by one animal may amount to as much as 100 m3/h (Magrin et al., 2017). The problem of excessive particulate pollution concerns not only livestock but also zoo animals and it worsens in autumn and winter months as animals from tropical and subtropical climates spend most of that time inside buildings. Well-being of these animals depends heavily on microclimate inside buildings where they stay (Veasey, 2017). By disturbing reproductive system airborne particle pollutants can decease the number of healthy offspring (Selevan et al., 2000; Perera et al., 2002), and as a consequence severely hamper or even prevent the process of saving threatened species—one of the main tasks undertaken by zoological gardens.

Available literature does not contain articles dealing with the influence of smog on the microclimate of buildings intended for animals, especially when it comes to zoological gardens. Existing research focuses on the impact of the particles generated as a result of animal production on employes handling animals or involved in degradation of the external environment (Mołocznik, 2004; Hulin et al., 2015).

As far as the impact of particulate matter on animals kept in the zoos is concerned, careful attention should be focused on antelopes as their high oxygen efficiency and lung capacity make them especially vulnerable to particulate pollution (Hinchcliff, Geor & Kaneps, 2008; Stafford & Stafford, 1993).

This article is concerned with assessing the impact of particulate air pollution on the concentration of PM10 and PM2.5 in the antelope house located in the Silesian Zoological Garden in Chorzów.

Materials and Methods

The research was conducted during the coldest period in 2018, that is between 27.02.2018 and 16.05.2018 (79 days) in the building housing antelopes located in the Silesian Zoological Garden in Chorzów (Fig. 1). The Zoological Garden is located on the border of Chorzów (108 thousand citizens) and Katowice (302 thousand citizens). This object constitutes a part of the Silesian Park. It is surrounded with forest complexes from the north and east. According to the records taken between 2014 and 2017 by the monitoring station of the Voivodship Inspectorate of Environmental Protection, located 2,400 m away from the Zoological Garden, there were 290 days when the limit value for PM10 was exceeded (Report, 2018).

Figure 1 A fragment of the orthophotomap showing developments in the immediate vicinity of the antelope house.

Based on the website www.geoportal.gov.pl.

Field experiments were approved by Silesian Zoological Garden in Chorzów (agreement number C/21/2013/WHiBZ). During taking measurements the building was occupied by 19 antelopes: Defassa waterbucks (Kobus ellipsiprymnus)—7 pieces weighing between 112 and 142 kg, addaxes (Addax nasomaculatus)—5 pieces with the weight ranging from 85 to 118 kg and lowland nyalas (Tragelaphus angasii)—7 pieces weighing 65–120 kg. Throughout the entire experimental period the animals stayed indoors, inside the boxes, with the possibility to move to the adjacent box only. Cleaning and litter replacement occurred once daily in the morning between 7:00 and 9:00.

Cubic volume of that building amounts to 3,626 m3, maximum height to 5.2 m and minimum height to 3.7 m. Total floor space of boxes amounts to 319 m2 (floor space of a single box for boxes no. 1–7 amounted to 11.5 m2, while for boxes no. 8–20 to 25 m2; Figs. 2 and 3). Animal boxes are provided with rubber flooring covered with long wheat straw with approximately 8% moisture content. The floor in the areas used by the visitors and passageways is covered with industrial flooring. The building is equipped with a mechanical supply-exhaust ventilation system, type Golem 4 with the capacity of 10,000 m3 per hour with class F6 filters. Specific fan power of supply air fans amounted to 1.33 kW/m3/s, while SFP for exhaust air fans to 1.02 kW/m3/s. Air flow rate in the supply air ducts reached 2.46 m/s, while in the exhaust air ducts 1.94 m/s. Air circulation in the studied building was assessed based on air flow measurements taken using a device Testo 425 (Testo Polska, Pruszków, Poland; resolution 0.01 m/s, accuracy 0.03 m/s).

Figure 2 Building plan for antelopes.

Figure 3 View of box (no. 15) with antelope.

Existing ventilation system provides only preliminary air filtration through a pocket filter (G4 592 × 592 × 360 mm 6k/metal), that filtrates the fractions with the size exceeding 10 um with 70% efficiency. Therefore, 30% of the particulate matter sized 10 um migrates inside the building and the fractions with smaller diameter are filtrated to a lesser degree (negligible scope). The ventilation operates 24/7. The ventilation system was not cleaned during taking measurements. The ventilation is cleaned twice yearly in the first week of February and July.

The study covered the following measurements: air temperature, relative air humidity, air pressure as well as the concentration of PM10 and PM2.5. The measurements were taken using two university measuring stations (UMS, serial no. U32 and U33, Poland) developed within the framework of the Storm & DustNet project implemented at the Jagiellonian University in Krakow. One measuring instrument was installed inside the studied building 2 m above the flooring, while the second was fitted outside 2 m above the ground at the distance of 30 m from the studied facility (Figs. 1 and 2). The device is fitted with a laser detector SEN0177 (DFRobot, Shanghai, China) that enables measuring the concentration of PM10 and PM2.5 in the air. The measurements of particle concentration and other parameters are taken continuously and the records of average values are compiled every minute. The accuracy of the particulate matter detector was verified by comparing data recorded during previous test measurements with the readings obtained from the reference station EDM107 belonging to GRIM company (Grimm & Eatough, 2009). The measurement error for EDM107 station amounts to ±2 µg/m3. This device has obtained a certificate of calibration and equivalency to a gravimetric method (Grimm & Eatough, 2009). Comparative measurements included natural ambient air and various particulate matter concentrations. The comparison revealed that measurements for PM2.5 and PM10 in UMS stations are encumbered with an error that does not exceed ±9 µg/m3. Measurements of the air temperature, humidity and pressure were taken using a BME detector fitted into the UMS station (temperature—measuring range from −40 to +85 °C, accuracy: ±1 °C; humidity—measuring range from 10% to 80% RH; accuracy: ±3% RH; air pressure—measuring range from 300 to 1,100 hPa, accuracy: ±1 hPa).

UMS stations measure PM concentrations and other parameters several dozen times per minute. Then the average minute values are computed. Later minute values are sent to the server and saved to a database. In the subsequent analyses average minute values are the basis for calculating average hourly and daily values.

Statistical analyses

Statistical analyses were performed using OriginPro 2016 software (OriginLab Corporation, Northampton, MA, USA). Normality distribution for variables was tested using the Shapiro–Wilk test. Due to the fact that obtained measurement results have no normal distribution, they were analyzed employing Spearman’s rank correlation coefficient (RS) with determining significance level (p-value). A 2-tailed test of significance was used in all studied cases.

Results

Air flow rate measured in the antelope house during the experiment ranged between 0.14 m/s and 0.18 m/s.

Daily average concentrations for PM10 and PM2.5 were determined based on particulate matter measurements. Distribution of these PM levels is shown in Figs. 4 and 5, respectively.

Figure 4 Distribution of daily average concentration for PM10 determined based on measurements taken inside (black) and outside (red) the antelope house in the period from 27.02.2018 to 16.05.2018.

Figure 5 Distribution of daily average concentration for PM2.5 determined based on measurements taken inside (black) and outside (red) the antelope house in the period from 27.02.2018 to 16.05.2018.

The measurements indicated that daily average concentration of 50 µg/m3 outside the building within the studied period was exceeded during 26 days.

The highest daily average concentration for PM10 in the air was recorded on 6th March 2018 (273 µg/m3).

Measurements taken inside the building indicted that in the studied period there were 11 days when the level of 50 µg/m3 was exceeded. The highest indoor daily average for PM10 was also recorded on 6th March and it amounted to 118 µg/m3. The concentration for PM10 outside the building was almost always higher than the one recorded indoors. We documented only two instances (19th April and 7th May) when the quantity of particulate matter inside the antelope house was higher by 1 µg/m3 than outside. However, it must be pointed out that these differences are significantly lower that the measurement error. Spearman’s correlation coefficient (RS) computed for the entire study period indicated a strong correlation (p-value < 0.01) between the concentration of PM10 inside and outside the building (RS = +0.91).

The distribution of daily average concentrations for PM2.5 for the studied period is shown in Fig. 3. According the measurements, in the studied period there where 35 days when the level of particulate matter outside the building exceeded 25 µg/m3 and 24 days when it comes to records taken indoors. The highest daily average concentration for PM2.5 was reached on 6th March (245 µg/m3 outside the building and 100 µg/m3 indoors). The concentration of PM2.5 inside the building has never exceeded the level recorded in the ambient air. Statistical analysis (RS = +0.93) revealed a strong correlation (p-value < 0.01) between the concentration of PM2.5 in the atmospheric air and inside the antelope house.

Statistical analysis showed a strong correlation between the concentration of PM10 and PM2.5 both inside and outside the building.

Spearman’s correlation coefficient (RS) between PM10 and PM2.5 outside and inside are large and indicate full dependance.

Calculation of the ratio for PM2.5 to PM10 concentration inside and outside the antelope house resulted in obtaining very similar values (0.84 and 0.85, respectively).

Daily reduction factor (RF) for the entire research period for PM10 was calculated according to Eq. (1), and the distribution in time for this factor is shown in Fig. 6.

(1) RF=PMoutdoor−PMindoorPMoutdoor×100

The calculations revealed that high value for reduction factor (RF) occurred between 1st and 10th as well as between 13th and 19th March and ranged between 50% and 65%, while on 26th April it reached 86%.

Figure 6 Distribution of daily reduction factor (RF) for PM10 calculated based on measurements taken inside and outside the antelope house in the period from 27.02.2018 to 16.05.2018.

Figures 7 and 8 illustrate average daily distribution of hourly values for PM10. In the case of Fig. 7 the analysis was conducted for the period with high PM content in the atmospheric air (27.02.2018–03.04.2018). The Fig. 8 refers to the period when the level of particulate matter in the air was low (04.04.2018–16.05.2018).

Figure 7 Distribution of daily average PM concentration inside (black) and outside (red) the antelope house based on hourly average for PM10 recorded from 27.02.2018 to 03.04.2018, that is in the period of high PM concentration in the atmospheric air.

Figure 8 Distribution of daily average PM concentration inside (black) and outside (red) the antelope house based on hourly average for PM10 recorded from 04.04.2018 to 16.05.2018, that is in the period of low PM concentration in the atmospheric air.

It has been observed that both in the case of high (Fig. 7) and low (Fig. 8) PM content in the atmospheric air, the concentration of particulate matter inside and outside the building followed the same pattern. The only differences were recorded between 7:00 and 9:00.

Analysis of the high frequency data—1 min (Fig. 9) revealed that the change in the PM concentration outside the building resulted in the shift in PM concentration indoors that was delayed only by 14 min. This delay was identified during establishing the maximum Spearman’s correlation coefficient as a function of time shift between PM10 distribution measured indoors and outdoors. There is no doubt that it was caused by a mechanical air supply system that operates round-the-clock. As shown in Fig. 9, the distribution of PM concentration outdoors (red line) is irregular and changeable, while the one for indoor PM concentration (black line) is more regular. Irregular distribution of PM outdoors is connected with the changes in air direction and flow rate (wind).

Figure 9 Distribution of PM10 (1 min resolution) determined based on measurements taken inside (black) and outside (red) the antelope house in the period from 03.03.2018 to 04.03.2018.

Daily average temperature outside the building ranged from −5 °C to 27 °C, while the temperature inside the antelope house was in the range of 17 °C to 27 °C. The distribution of daily average temperatures in described period is shown in Fig. 10.

Figure 10 Distribution of daily average temperatures determined based on measurements taken inside (black) and outside (red) the antelope house in the period from 27.02.2018 to 16.05.2018.

Statistical analysis revealed a strong correlation between average hourly temperatures outside and inside the studied building (RS = +0.88). Relative air humidity measured outside the building reached the values from 39% to 68%, while inside it ranged from 20% to 52% (Fig. 11).

Figure 11 Distribution of daily average relative humidity determined based on measurements taken inside (black) and outside (red) the antelope house in the period from 27.02.2018 to 16.05.2018.

Atmospheric pressure in the studied period ranged between 965 hPa and 1,000 hPa (Fig. 12).

Figure 12 Distribution of daily average air pressure determined based on measurements taken in the period from 27.02.2018 to 16.05.2018.

Correlation coefficient between average hourly humidity inside the antelope house and humidity in the ambient air amounted to RS = +0.40.

The calculations indicated a strong negative correlation between thermal conditions outside the antelope house and the concentration for PM2.5 and PM10 inside the building (RS: −0.62 and −0.63, respectively). We discovered a weak correlation between relative humidity outdoors and the concentration of PM10 and PM2.5 inside the building (RS: +0.35 and +0.39, respectively), as well as a weak correlation between atmospheric pressure and the concentration for PM2.5 and PM10 inside the antelope house (RS = −0.31).

Discussion

The basic function of a building is to provide people and animals with as good as possible living conditions. Buildings intended for animals shall be equipped with a ventilation system that removes used air and supplies atmospheric air (Collins, 1990). When the air outside is heavily polluted, the ventilation, despite improving the quality of the air inside the building, contributes to increasing pollutant levels indoors (Qi et al., 2017). The measurements taken in the course of the study indicated that during the period of high dust loading in the ambient air, high concentration of particulate matter also occurs inside the antelope house. Unfortunately, available materials fail to provide standards for permissible exposure to particulate matter inside buildings intended for animals. When it comes to people matters related to standards for particulate matter concentration are governed by the Directive CAFE (CAFE, 2008). But in the case of animals we can base our assumptions only on the information included in the materials issued by the European Council regarding well-being, stating that the concentration of particulate matter in the buildings where the animals are housed must be maintained on the level that is not harmful to animals (Council of the European Union, 1998). As far as legislation in Poland is concerned, standards for the maximum permitted PM concentration for animals have not been established yet. For people the average 24-h PM10 concentration cannot exceed 50 μg/m3 (https://powietrze.gios.gov.pl/pjp/content/annual_assessment_air_acceptable_level).

For that reason, in this article we have used standards established for people. In the studied period we recorded 11 days when PM10 concentration exceeded 50 µg/m3 and 24 days when PM2.5 was higher than 25 µg/m3 inside the antelope house.

The measurements taken indicated that despite fitting air filters in the ventilation system, particulate matter from the atmospheric air migrated into the building and the correlation coefficient between the concentration of particles indoors and outdoors was statistically significant. The building with its ventilation system reduced PM10 concentration by approximately 60% during the highest smog level as compared with the ambient air. Despite considerable reduction, the concentration of particulate matter inside the building exceeded the standard established for people (CAFE, 2008) more than twofold. The research conducted by Wenke et al. (2018) inside the building intended for animals with different ventilation systems and filter types demonstrated that the difference between particulate matter concentration in rooms with and without filters amounts to 10 ± g/m3 on average. Challoner & Gill (2014) and Massey et al. (2009) report that efficient reduction of particulate content by means of mechanical or natural ventilation in buildings intended for people is not possible.

The increase of PM concentration between 7:00 and 9:00 (Fig. 7) as well as 7:00 and 8:00 (Fig. 8) was most probably associated with cleaning and replacing litter, what resulted in additional PM increment inside the building.

The number of particles transported with the air inside the respiratory system depends on, among others, the volume of the air inhaled by the animal, that is pulmonary capacity reduced by so-called respiratory dead space. For the Defassa waterbucks the volume of inhaled air amounts to approximately 9 dm3, for addaxes 6 dm3 and 4 dm3 for lowland nyalas (Muir & Hubbell, 2012). Assuming that the average number of breaths per minute for these animals in resting state amounts to 25, we are able to estimate that in the period with the highest particle concentration (PM10 = 118 µg/m3, PM2.5 = 100 µg/m3) the antelopes introduced into the respiratory system approximately 1,062 µg of PM10 including 900 µg of PM2.5. It should be noted that these values are accurate only if we assume that animals were in resting state. Undertaking physical activities is associated with increasing the volume of inhaled air (increased depth of breathing and the number of breaths per minute).

During breathing PM2.5–10 migrate into the tracheal and bronchial part of the respiratory system, while PM2.5 reach lungs and through the cardiovascular system may be distributed throughout the body (Wei et al., 2018; WHO, 2013). The measurements taken in the building under the study revealed that as much as 85% of the particulate mass constitutes PM2.5. Migration of the particulate matter inside the antelope house is also associated with introducing other substances, among others, benzo(a)pyrene (BaP), lead (Pb), arsenic (As), cadmium (Cd), nickel (Ni) (Wróbel, Rokita & Maenhaut, 2000). Taking into account the average concentration of these substances in the ambient air in the period with the heaviest smog emission (Report, 2018) as well as the reduction of the particulate matter quantity by the building, it is possible to calculate that the antelopes inside the facility introduced to their bodies through the respiratory system: 94.59 ng/m3 BaP, 0.18 µg/m3 Pb, 10.62 ng/m3 As, 8.91 ng/m3 Cd and 4.5 ng/m3 Ni per hour.

The negative correlation between the outside air temperature and the concentration of particulate matter indoors indicates that a decrease of temperature is followed by an increase in the quantity of burned fossil fuels used to heat houses, what in turn has influence on the level of particles in the atmospheric air. Particles generated in that way are transported through the ventilation system inside the buildings.

The discussion on the air pollution inside the antelope house should also cover the aspects connected with people involved in handling animals. During 8-h working day at the highest particulate concentration these people inhale on average approximately 1,173 µg of PM10, including 972 µg of PM2.5 (Pocock, Richards & Richards, 2013).

Trees are capable of reducing particulate matter concentration (Sæbø et al., 2012). Despite the fact that the antelope house is surrounded by forest complexes filtering polluted air, the amount of PM produced by two neighboring towns (Chorzów and Katowice) is so large that it is impossible for the plants growing around the antelope house to efficiently clean the air (>50 µg/m3) in the vicinity of the building (Fig. 4, red line).

Conclusions

The experiment revealed that the quantity of particulate matter inside the building intended for antelopes is strongly correlated with the concentration of particles in the atmospheric air. In spite of the fact that the existing ventilation system is equipped with a reduction filter that lowers the dust level by 60% during the heaviest smog, particulate matter concentration in the antelope house exceeded acceptable limit for PM10 more than twofold. Particle size-fraction analysis revealed that as much as 85% of the particles identified in the studied building constitute PM2.5. These particles are capable of penetrating into the lungs causing its irritation or damage, as well as infiltrate through pulmonary alveolars into the vascular system. These results indicate that it is necessary to install suitable filters in the air supply ducts that would efficiently reduce the level of PM in air supplied from outside. The results obtained in the course of this study also indicate a strong need to conduct further research into the quality of air inside the facilities intended for animals as well as necessity to establish permissible particulate matter concentrations for farm, exotic and household animals.

Supplemental Information

Supplemental Information 1 Concentration dust PM10, air temperature, relative humidity, pressure inside of building.

Click here for additional data file.

Supplemental Information 2 Concentration dust PM10, air temperature, relative humidity, pressure outside of building.

Click here for additional data file.

Supplemental Information 3 Concentration dust PM2.5, air temperature, relative humidity, pressure inside of building.

Click here for additional data file.

Supplemental Information 4 Concentration dust PM2.5, air temperature, relative humidity, pressure outside of building.

Click here for additional data file.

Supplemental Information 5 Raw data (time resolution 1 minute) recorded during 03.03.2018 and 04.03.2018 by station U33 outside building, applied for analyses and presented in Figure 9 (line color red).

Click here for additional data file.

Supplemental Information 6 Raw data (time resolution 1 minute) recorded during 03.03.2018 and 04.03.2018 by station U32 inside building, applied for analyses and presented in Figure 9 (line color black).

Click here for additional data file.

Additional Information and Declarations

Competing Interests

Author Contributions

Field Study Permissions

Data Availability

The authors declare that they have no competing interests.

Krzysztof Pawlak conceived and designed the experiments, performed the experiments, analyzed the data, prepared figures and/or tables, authored or reviewed drafts of the paper, and approved the final draft.

Zenon Nieckarz performed the experiments, analyzed the data, prepared figures and/or tables, authored or reviewed drafts of the paper, and approved the final draft.

The following information was supplied relating to field study approvals (i.e., approving body and any reference numbers):

Field experiments were approved by Silesian Zoological Garden in Chorzów (agreement number C/21/2013/WHiBZ).

The following information was supplied regarding data availability:

The raw measurements are available in the Supplemental Files.

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
