# Peer review of "The impact of smog on the concentration of particulate matter in the antelope house in the Silesian zoological garden"

_PeerJ, doi:10.7717/peerj.9191_

## Round 0.1 · original submission · Major Revisions

Whilst in my opinion the experimental work may be publishable, the current manuscript requires additional information to improve the utility of the paper. These revisions are based on my opinion as editor and this paper has not yet been assigned reviewers.When the following changes have been made the paper will be assigned reviewers.

1. There is a lack of fine detail describing the management of the building environment. I would like to see as much contextual information as possible and at an appropriate timescale. For example, how is the HVAC system controlled? Was the building constantly ventilated? Do you have a time series of operation of the ventilation system. The operation of the heating and ventilation system will directly impact on the dilution of indoor sources of PM and the ingress of outdoor PM. Does the ventilation system recycle indoor air through the PM filters? Was the system recently maintained?

2. The paper does not consider the role of PM sources within the building. Housing animals within a buildings i likely to cause the production of PM within the building itself. Was there a cleaning regime? What was it and how often was it performed? Do the animals exercise regularly? How and when? Moving animals in a building will likely be a PM source.

3. The daily averages are not sufficient especially when considering that the OPCs will give high frequency data. More details of the sensor and the sensor placement needs to be given. A figure illustrating the location of the indoor and outdoor measurement locations and the geometry of the building should be included. The local meteorology impacting the site should be presented and used to provide context to the analysis. Higher frequency data will allow you to determine the influence of various sources and may allow the differentiation between indoor and outdoor components. Examples of analysis might be diel averages of measurements to compare outdoor and indoor variability at finer scale than one day.

---

## Round 0.2 · Major Revisions

Please respond as well as you can to the comments of both the reviewers. I think it is especially important to give credit the previous work on air quality that already exists in the literature and improve the support for assertion made with citations.

Improvements to the time resolution of the data analysis will also provide stronger support for the conclusions of the paper which suggests that outdoor air is determining the indoor composition.

Reviewer 1 ·

Basic reporting

The paper is written in a good English overall and it can be followed and understood.
I think there is a bit of lack of references especially in the introduction (Line from 33 to 37) where statements are made but there is no reference to support the statement (for as obvious as they might sound a reference is always good). Same issue with the sentence from line 51 to 53 in the introduction.

Experimental design

The research question is not very well defined. Although i can understand that it is important for the animal in a zoo to be kept in a good shelter, why is this study specifically focusing on these animals? Why not checking other animals? Are these animals more prone to lungs problems?

The answers to the editor are not complete. Also in my opinion more information on the circulation of the air in the building should be given and what was answered currently to the points of the editor are far from being satisfactory.

Why are high frequency data still not showed? it is not clear if those data are not available because the provided one are already averaged or why the decision was to use daily average data. Maybe it does not make any difference but it is not possible to judge without checking the data.

Validity of the findings

i would say that without the addition of what the editor already requested in more details it might be difficult to asses the impact of the findings. Assuming it is correct that the particulate indoor is affected by the particulate outdoor, it is not clear how this is a larger issue for the animals compared to, for example, being locked in small boxes etc...Are the authors trying to make an analogy with humans leaving indoor and also needing to worry about quality of the air?

Additional comments

I would suggest to strongly consider in a better way the comments from the editor and what was written above. The message of why the animals (and these specific animals) need to be done better as at the moment it is not clear.

·

Basic reporting

There are some letter mistakes, some of sentences are not fully clear. I suggest to read it one more time carefully or give for correction to native speaker.

Experimental design

Methods and experimental design correct.

Validity of the findings

The review concerns the work on impact of smog on the concentration of particulate matter in the antelope house in the Silesian zoological garden. The issue raised at work is new. During my work on air pollution, I have not yet come across the impact of dust pollution on animals and rooms in zoos. At the same time, as the authors describe, this problem can be very important. That is why I believe that the work as one of the first on this subject may initiate a new field of research on PM pollution and their impact.

Additional comments

Comments:
Line 22 - Why authors did research on this specific time period. Why not through the whole year?
Line 23 - and other PM10 we are writing with subscript.
Line 30 - and other PM2.5 we are writing with subscript.
Line 33 - delete so-called
Line 34 - Dust its not a proper word. You are writing about Particulate matter pollution (PM), So this words should be used. PM is scientific name of this type of pollution.
Line 37. Fine PM or fine particulate matter particles.
Line 47 - unneeded and on the beginning
Line 53 - Authors should explain how the norms in Poland look like
Line 77 - I don't think the time period between March and Mai is the coldest time in Poland.
Line 76. There is a miss of information about the city of Chorzów. About the particulate pollution in the city, and about in which part of the city the zoological garden is situated. Suburbs or center.
Line 81. The Latin name of the species and author of the name is missing
Line 88. 11,4 m2 should be in superscript.
Line 101. The brand of the station is missing
Line 105. The brand of laser
Line 111 and 115. m3 should be in superscript.
Line 131. The sentences should aligned.

In the whole work there is a miss of role of plants - trees and shrubs in decreasing the amount of PM in air. A lot works were done even for polish cities. Chorzów zoological garden is full of trees. It should be mentioned even in one sentence in the introduction part as a citation. Moreover plants inside the buildings can reduce ale the pollution, we don't know if they are some inside antelope building. Im not specialist of the animals but maybe some should be placed there.

---

## Round 0.3 · accepted · Accept

Thank you for for engaging with the comments of the reviewers. The reviewer is satisfied with how you addressed their comments and suggestions.

Reviewer 1 ·

Basic reporting

no comment

Experimental design

no comment

Validity of the findings

no comment